# Synthesis of Green-Emitting Gd_2_O_2_S:Pr^3+^ Phosphor Nanoparticles and Fabrication of Translucent Gd_2_O_2_S:Pr^3+^ Scintillation Ceramics

**DOI:** 10.3390/nano10091639

**Published:** 2020-08-20

**Authors:** Zhigang Sun, Bin Lu, Guiping Ren, Hongbing Chen

**Affiliations:** 1School of Materials Science and Chemical Engineering, Ningbo University, Ningbo 315211, China; 273495951@qq.com (Z.S.); chenhongbing@nbu.edu.cn (H.C.); 2Key Laboratory of Photoelectric Materials and Devices of Zhejiang Province, Ningbo 315211, China; 3Faculty of Maritime and Transportation, Ningbo University, Ningbo 315211, China; 404603682@qq.com

**Keywords:** Gd_2_O_2_S, Pr^3+^ doping, ceramic scintillator, phosphor powder

## Abstract

A translucent Gd_2_O_2_S:Pr ceramic scintillator with an in-line transmittance of ~31% at 512 nm was successfully fabricated by argon-controlled sintering. The starting precipitation precursor was obtained by a chemical precipitation route at 80 °C using ammonia solution as the precipitate, followed by reduction at 1000 °C under flowing hydrogen to produce a sphere-like Gd_2_O_2_S:Pr powder with an average particle size of ~95 nm. The Gd_2_O_2_S:Pr phosphor particle exhibits the characteristic green emission from ^3^P_0,1_→^3^H_4_ transitions of Pr^3+^ at 512 nm upon UV excitation into a broad excitation band at 285–335 nm arising from 4*f*^2^→4*f*5*d* transition of Pr^3+^. Increasing Pr^3+^ concentrations induce two redshifts for the two band centers of 4*f*^2^→4*f*5*d* transition and lattice absorption on photoluminescence excitation spectra. The optimum concentration of Pr^3+^ is 0.5 at.%, and the luminescence quenching type is dominated by exchange interaction. The X-ray excited luminescence spectrum of the Gd_2_O_2_S:Pr ceramic is similar to the photoluminescence behavior of its particle. The phosphor powder and the ceramic scintillator have similar lifetimes of 2.93–2.99 μs, while the bulk material has rather higher external quantum efficiency (~37.8%) than the powder form (~27.2%).

## 1. Introduction

Scintillation materials convert the radiation of high-energy rays (X-rays or gamma rays) into visible light and are extensively applied in various fields such as safety inspection, high energy physics, nuclear medicine, and industrial non-destructive testing [1,2,3,4,5,6]. Scintillators can be divided into solid, liquid, and gaseous states, among which solid inorganic scintillators, as the most widely used materials, include single crystals and polycrystalline ceramics. A ceramic scintillator is superior to a single crystal due to its low cost, short production cycle, large-size production, as well as high dopant concentration with a homogeneous mixture at the molecular level [6,7,8,9]. Solid-state Gd_2_O_2_S (GOS) has excellent chemical and physical characteristics, such as high melting point (2070 °C), high density (7.43 g/cm^3^), high X-ray attenuation coefficient (~52 cm^−1^ at 70 keV), wide band gap (4.6–4.8 eV), favorable chemical durability, low phonon energy, low crystal symmetry, and low toxicity, which make it a promising host material for luminescence and scintillation applications [10,11]. GOS-based phosphors have attracted considerable attention for decades. For instance, the green-emitting GOS:Tb phosphor exhibits a high brightness and a high luminous efficiency and is applied in television screens, cathode ray tubes, and X-ray intensifying screens [12,13].

GOS belongs to the hexagonal crystal structure (space group: $P3^{-}$
*ml*; lattice parameters: *a* = *b* = 0.3851 nm and *c* = 0.6664 nm) with trigonal symmetry [10]. Owing to its optically anisotropic structure, the sintered GOS ceramic is difficult to densify into a fully transparent body by sintering densification. The obtained GOS bulk generally shows an opaque or a translucent appearance, which mainly depends on its relative density. The Pr^3+^-activated GOS ceramic is an effective ceramic scintillation material for high-quality imaging applications owing to its high X-ray conversion efficiency, strong stopping power for the incident X-ray, short lifetime, fast afterglow, high chemical stability, and good emitting, matching with the high sensitivity of silicon photodiodes. Furthermore, co-doping with Ce^3+^ or F^−^ was found to have significant impacts on the scintillation properties, such as light output and afterglow [8,14,15].

GOS-based ceramic scintillators have been fabricated by hot pressing (HP) [14,16,17], hot isostatic pressing (HIP) [18], spark plasma sintering [19], and pressureless sintering in hydrogen/nitrogen atmosphere [20,21]. Although HP/HIP sintering has low requirements in terms of particle sinterability, the disadvantages lie in low efficiency and expensive cost. On the other hand, pressureless sintering is cost-efficient and time-effective, but it requires sinterable starting powder. 

To date, oxysulfide particles have been synthesized by three main paths, namely, solid phase reaction, gaseous sulfuration method, and liquid phase route [22,23,24,25,26,27,28,29,30]. The traditional solid-state reaction frequently needs ball-milling treatment and high reaction temperature, leading to coarse particles, uneven particle size distribution, and undesired contamination [31,32]. The gaseous sulfuration method results in superfine GOS powders, but harmful vapors such as H_2_S, CS_2_, or S can be simultaneously produced [30]. The wet chemical route is an environmentally friendly way for the synthesis of morphology-controllable oxysulfide particles, including hydrothermal reaction [20,33], homogenous precipitation [6,34,35,36,37], and direct precipitation [38], among which direct precipitation is preferred for its higher batch yield, time saving, and simple operation.

In the present work, nanocrystalline Gd_2_O_2_S:Pr powder was obtained by a reduction reaction under a hydrogen atmosphere at 1000 °C from its precipitation precursor prepared by a direct precipitation method, with which translucent Gd_2_O_2_S:Pr ceramic was successfully fabricated by pressureless sintering under a protective argon atmosphere. The luminescence and scintillation properties of both the Gd_2_O_2_S:Pr phosphor and ceramic were studied in detail.

## 2. Materials and Methods

### 2.1. Particle Synthesis 

The starting materials of Gd_2_(SO_4_)_3_∙6H_2_O (>99.99% pure; Haoke Technology Co., Ltd., Beijing, China) and Pr_2_(SO_4_)_3_∙6H_2_O (>99.99% pure; Haoke Technology Co., Ltd., Beijing, China) were together dissolved in distilled water according to the stoichiometric ratio of (Gd_1-*x*_Pr*_x_*)_2_O_2_S (*x* = 0.001–0.0075) and well mixed to make a homogeneous mother liquor with a concentration of 0.01 mol/L.

A white suspension was yielded by dropwise dripping 1.0 mol/L NH_3_∙H_2_O (GR pure, Aladdin Biochemical Technology Co., Ltd., Shanghai, China) into the mother liquor in a hot-water bath of 80 °C at a rate of 3–5 mL/min under magnetic stirring (~180 r/min) until pH = 6.5. After aging for 1 h, the white precipitation was repeatedly rinsed with distilled water via centrifugal separation. The precipitation precursor was then dried at 100 °C for 12 h and lightly ground with an agate mortar. The precursor was calcined at 1000 °C for 3 h in a tubular furnace under flowing hydrogen (~150 mL/min) to yield oxysulfide.

### 2.2. Ceramic Production

The as-prepared oxysulfide powder was compressed by cold isostatic pressing under ~240 MPa. The green body was sintered in a tungsten-heater furnace at 1650 °C under protective argon atmosphere. The density of the sintered body was determined by Archimedes’ method. The sintered ceramics were double-side ground and polished to be ~11.15 ± 0.01 mm in diameter and ~0.50 ± 0.01 mm in thickness.

### 2.3. Characterization

The precipitation precursor and its reduction product were characterized by field-emission scanning electron microscopy (FE-SEM; Model S-4800, Hitachi, Japan) and X-ray diffraction (XRD; Model D8 Focus, Bruker, Germany) using nickel-filtered CuK*α* as the incident radiation. The surface microstructure of the ceramic was observed on a desktop scanning electron microscope (SEM; Model EM-30plus, COXEM, Korea). Photoluminescence (PL)/photoluminescence excitation (PLE) spectrum, fluorescence lifetime, and quantum yield of the powder and ceramic were measured by transient fluorescence spectrophotometer (Model FLS 980, Edinburgh Instruments Ltd., Livingston, UK) using a 450 W xenon lamp as the excitation source. The X-ray excited luminescence (XEL) spectrum of the ceramic was measured using the photomultiplier tube working on a Zolix Omni-λ300 monochromator at a voltage of −900 V, while the X-ray tube copper target was operated at a voltage of 69 kV and a current of 3 mA.

## 3. Results and Discussion

Figure 1A shows the XRD patterns of the precursor powder and reduction products. An amorphous precursor was obtained via the chemical precipitation route. The reaction temperature of 80 °C with an aging time of 1 h is not enough for its crystallization. Upon calcination at 1000 °C under a hydrogen atmosphere, all diffraction peaks of the products can be well indexed to hexagonal Gd_2_O_2_S (JCPDS No. 26–1422) without any impure phases, indicating that the precursor fully converts into oxysulfide via thermal decomposition and reduction reaction. Its schematic crystal structure drawn by VESTA 3D visualization software is shown in Figure 1B. Only one coordination type for Gd^3+^ exists in the unit cell, that is, each Gd^3+^ is surrounded by four oxygen atoms and three sulfur atoms to form a seven-coordination polyhedron [39]. After Pr^3+^ incorporation, it would substitute the Gd^3+^ site because of their similar ionic radii (0.0990 nm for Pr^3+^ with CN = 6 and 0.0938 nm for Gd^3+^ with CN = 6). Along with increasing Pr^3+^ addition, remarkable shifts of diffraction peaks were not observed due to the low Pr^3+^ concentration and small difference in ionic radii between Pr^3+^ and Gd^3+^. The average crystallite size (*D_XRD_*) can be calculated by the Debye–Scherrer formula: *d* = 0.89*λ*/[(*B*_0_^2^ − *B_c_*^2^)^1/2^*∙cosθ*], where *B*_0_ is the half-peak width, *B_c_* is the correction factor caused by instrument broadening, *θ* is the angle of the diffraction peak, and *λ* is the wavelength of the X-ray [40]. The resulting *D_XRD_* values calculated from (101) diffractions are ~35.7, 32.3, 58.0, and 47.7 nm for the samples doped with 0.1, 0.25, 0.5, and 0.75 at.% Pr^3+^, respectively.

Figure 2 displays morphologies of the precipitation precursor and its reduction product at 1000 °C. The precipitation precursor shows a two-dimensional nanoplate-like shape clustered into a loose honeycomb. Although the XRD analysis cannot indentify the chemical composition of the precursor, it may be speculated to possess the layered Ln_2_(OH)_4_SO_4_∙*n*H_2_O (Ln = Gd and Pr) structure consisting of the host layer of Ln-hydroxy polyhedron and the interlayer SO_4_^2−^ [25,33], because this compound not only exhibits the nanoplate morphology but also has the same Ln/S molar ratio as the (Gd,Pr)_2_O_2_S reduction product (Figure 1). The precipitation formation mechanism obeys the hard-soft acid-base principle, viz., the hard acid tends to react with the hard base and the same for the soft acid and the soft base—hard-hard or soft-soft combinations [41]. The hard Lewis acid of Gd^3+^ is readily coupled with the hard Lewis bases of SO_4_^2−^ and OH^−^ to form the basic sulfate. After being reduced at 1000 °C, the nanoplate-like precursor is completely cracked into a sphere-like oxysulfide powder with a statistic average particle size of ~95 nm.

The appearance of the GOS:Pr ceramic fabricated by argon-controlled sintering is shown in Figure 3A. Although the ceramic sample exhibits translucence to the naked eye, it is a relatively good optical quality for a ceramic that has an optically anisotropic crystal structure. Under UV irradiation, the bulk emits the strong green visible light derived from ^3^P_1,0_→^3^H_4_ transitions of Pr^3+^. On the ceramic transmittance curve (Figure 3B), the absorption band at ~350 nm corresponds to the 4*f*^2^→4*f*5*d* transition of Pr^3+^ and the other ones beyond 350 nm are assignable to the intra−4*f*^2^ transitions of Pr^3+^. The GOS:Pr bulk has an in-line transmittance of ~31% at 512 nm (Pr^3+^ emission center). Figure 3C,D shows the surface microstructure and fracture surface of the sintered GOS:Pr ceramic. The specimen has a dense microstructure, and pores are only occasionally observed. Such a microstructure is desired for improved scintillation performance, since the pores frequently induce scattering losses. Its relative density was measured to be ~99.2% by Archimedes’ method. The statistic average grain size is ~5 μm via WinRoof image analysis software. The grain size is relatively uniform, and exaggerated grain growth is not found. By observing its fracture surface, the dense GOS:Pr ceramic is mainly intragranularly fractured.

Figure 4A presents the excitation spectra of (Gd_1-*x*_Pr*_x_*)_2_O_2_S (*x* = 0.001–0.0075) phosphor particles obtained by monitoring 512 nm emission of Pr^3+^ as a function of activator concentration. The main broad bands of all samples are located at 285–335 nm, which are ascribed to 4*f*^2^→4*f*5*d* transitions of Pr^3+^ [34]. With more Pr^3+^ addition, a ~2 nm redshift of the 4*f*^2^→4*f*5*d* transition band center can be observed. Such a phenomenon can be attributed to the lower electronegativity of Pr^3+^ (1.13) than that of Gd^3+^ (1.21), which allows for the easier electron transfer to the excitation state of Pr^3+^. The weak bands at 245–285 nm are assignable to the absorption of the host lattice, since the corresponding bandgap of Gd_2_O_2_S was found to be ~4.6 eV [34]. Redshift lattice-absorption bands (~6 nm) are also observed, because increasing Pr^3+^ incorporation would lead to a smaller bandgap for the oxysulfide. Under 306 nm excitation, all the (Gd_1-*x*_Pr*_x_*)_2_O_2_S phosphors exhibit characteristic green emissions of Pr^3+^ (Figure 4B). That is, the strongest emission peak at ~512 nm originates from the ^3^P_0_→^3^H_4_ transition; the second strongest peak at ~502 nm derives from the ^3^P_1_→^3^H_4_ transition; and the weakest peak at ~547 nm arises from the ^3^P_1_→^3^H_5_ transition. Both the PLE and PL intensities of (Gd_1−*x*_Pr*_x_*)_2_O_2_S phosphors increase to 2.6–2.8 times with the gradually rising Pr^3+^ concentrations from 0.1 to 0.5 at.% as shown from the normalization curves (Figure 4C,D). A further increase in Pr^3+^ content (e.g., 0.75 at.%), however, causes a reduction in PL/PLE intensity due to luminescence quenching. Thus, the optimal Pr^3+^ concentration in the GOS matrix is determined to be 0.5 at.%. As a function of the Pr^3+^ content, the PLE and PL intensities have a highly consistent variation tendency.

Huang et al. propose a theory to describe the relationship between PL intensity and activator concentration [42], which agrees with Dai and Meng et al. [43,44]. That is, the mutual interaction type of luminescence quenching in a solid phosphor can be determined by the following equation:(1)log(I/c)=(−s/d)log(c)+logf
where *I* is the emission intensity, *c* is the activator concentration, *s* is the index of electric multipole, *d* is the sample dimensionality (*d* = 3 for energy transfer among the activators inside particles), and *f* is the constant. The variable *s* values correspond to different quenching mechanisms. Namely, the *s* values of 6, 8, and 10 relate to the dipole–dipole, dipole–quadrupole, and quadrupole–quadrupole electric interactions, respectively, while *s* = 3 corresponds to exchange interaction. The plot of *log*(*I*/*c*) versus *log*(*c*) for the 512 nm emission of Pr^3+^ is shown in Figure 5, from which a linear slope (*s*/3) of ~0.7 is yielded and thus the *s* value is close to 3. It can be concluded that the exchange interaction is mainly responsible for the luminescence quenching of GOS:Pr phosphors. The exchange interaction processes are divided into radiative and non-radiative. The former includes emission and radiative transfer, whilst the latter comprises internal relaxation and multipolar interactions between ions. The PL intensity linearly rises as Pr^3+^ concentration increases up to 0.5 at.%, since more luminous centers are generated. However, a further increase in Pr^3+^ content (e.g., above 0.75 at.%) enhances the probability of energy transfer with cross-relaxation between Pr^3+^ activators due to the shortened distances.

Figure 6A shows the XEL spectrum of the translucent GOS:Pr ceramic scintillator made in this work. The ceramic material exhibits a strong green emission at 510–514 nm, arising from ^3^P_0_→^3^H_4_ transition of Pr^3+^, which is similar to the PL behavior of its powder form. The mechanisms between PL and XEL substantially differ from each other. The PL primarily utilizes the 4*f*^2^→4*f*5*d* transition of Pr^3+^. However, the XEL can be divided into the following three processes:(2)$$X-\mathrm{rays}\to e^{\_}+h^{+}$$
(3)$${\mathrm{Pr}}^{3+}+h^{+}\to {\mathrm{Pr}}^{4+}$$
(4)$${\mathrm{Pr}}^{4+}+e^{\_}\to ({\mathrm{Pr}}^{3+})*\to {\mathrm{Pr}}^{3+}+hv$$

That is, under high-energy ray excitation, lots of electron-hole pairs in the host lattice are created [45]. The Pr^3+^ cation traps the hole to form a transient Pr^4+^ state [15], followed by recombination with the electron to emit visible light. The 1931 CIE chromaticity coordinate of the GOS:Pr ceramic is (0.11, 0.73), which falls into the characteristic green region (Figure 6B).

Figure 7A exhibits the decay kinetics of the GOS:Pr phosphor powder and ceramic scintillator for the 512 nm emission of Pr^3+^ under 306 nm excitation. The fluorescence lifetime can be obtained via fitting the decay curve with the single exponential equation: *I*(*t*) = *Aexp*(−*t*/*τ*) + *B*, where *τ* is the fluorescence lifetime, *t* is the delay time, *I*(*t*) is the instantaneous emission intensity, and *A* and *B* are constants [45,46]. The fitting results yield *τ* = 2.93 ± 0.02 μs, *A* = 425.74 ± 1.09, and *B* = 10.12 ± 0.51 for the phosphor powder, and *τ* = 2.99 ± 0.03 μs, *A* = 264.73 ± 1.39, and *B* = 7.89 ± 0.38 for the scintillation ceramic. The fluorescence lifetimes determined in this work are in general agreement with the reported values of 2.4–3.0 μs for GOS:Pr,Ce ceramics [8,11,18,47].

Figure 7B exhibits the responses of the GOS:Pr phosphor powder and ceramic scintillator to 306 nm excitation using a white BaSO_4_ solid as a reference material. The external quantum efficiencies (*ε*_ex_) of the sample can be deduced from the total number of emitted photons divided by the total number of excited photons as follows [45,48]:(5)$$\varepsilon_{ex}=\frac{\int \lambda P(\lambda )d\lambda }{\int \lambda E(\lambda )d\lambda }$$
where *P*(*λ*)/*h**ν* and *E*(*λ*)/*hν* are the numbers of photons in the emission and excitation spectra of the samples, respectively. The reflection spectrum of the white standard is used for calibration. The external quantum efficiencies of the GOS:Pr powder and bulk are determined to be ~27.2% and 37.8%, respectively. The rather higher *ε*_ex_ for the latter is attributed to the improved crystallinity and rapid grain growth via high-temperature sintering.

## 4. Conclusions

A precipitation precursor with two-dimensional nanoplate-like morphology was prepared at 80 °C using ammonia as the precipitant, followed by reduction at 1000 °C under a hydrogen atmosphere to yield a hexagonal Gd_2_O_2_S:Pr phosphor powder. After cold isostatic pressing and argon-controlled sintering, the obtained Gd_2_O_2_S:Pr scintillation ceramic has a dense microstructure with a relative density of ~99.2%. The main conclusions from this work can be summarized as follows: (1) The sphere-like Gd_2_O_2_S:Pr phosphor powder has an average particle size of ~95 nm and exhibits the characteristic green emission from ^3^P_0,1_→^3^H_4_ transitions of Pr^3+^; (2) The optimum concentration of Pr^3+^ is 0.5 at.%, and the luminescence quenching type is dominated by exchange interaction; (3) The Gd_2_O_2_S:Pr ceramic has an in-line transmittance of ~31% at 512 nm upon X-ray excitation into the strong green emission with a 1931 CIE chromaticity coordinate of (0.11, 0.73); (4) The phosphor powder and the ceramic bulk have similar lifetimes of 2.93–2.99 μs; (5) The Gd_2_O_2_S:Pr ceramic scintillator has a higher external quantum efficiency (~37.78%) than the powder form (~27.2%).

## Figures and Tables

**Figure 1 nanomaterials-10-01639-f001:**
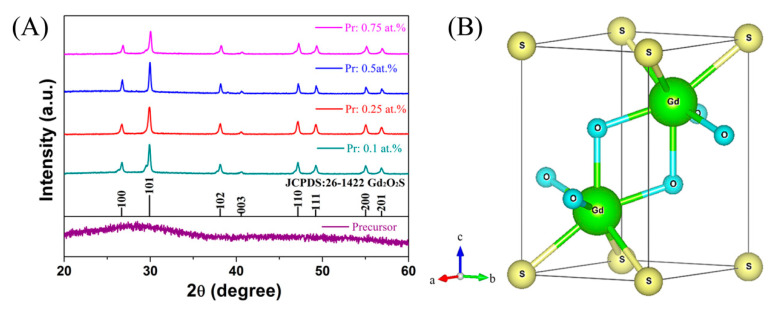
XRD patterns of the Gd_2_O_2_S (GOS):Pr precursor powder and reduction products (**A**), and schematic crystal structure of GOS (**B**).

**Figure 2 nanomaterials-10-01639-f002:**
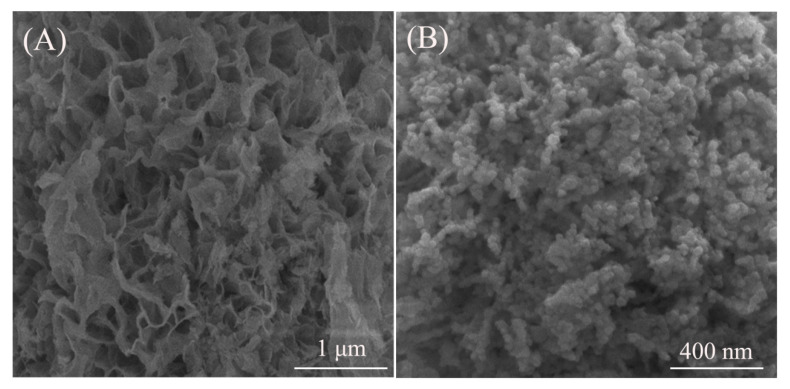
FE-SEM micrographs showing morphologies of the precipitation precursor (**A**) and its reduction product at 1000 °C (**B**).

**Figure 3 nanomaterials-10-01639-f003:**
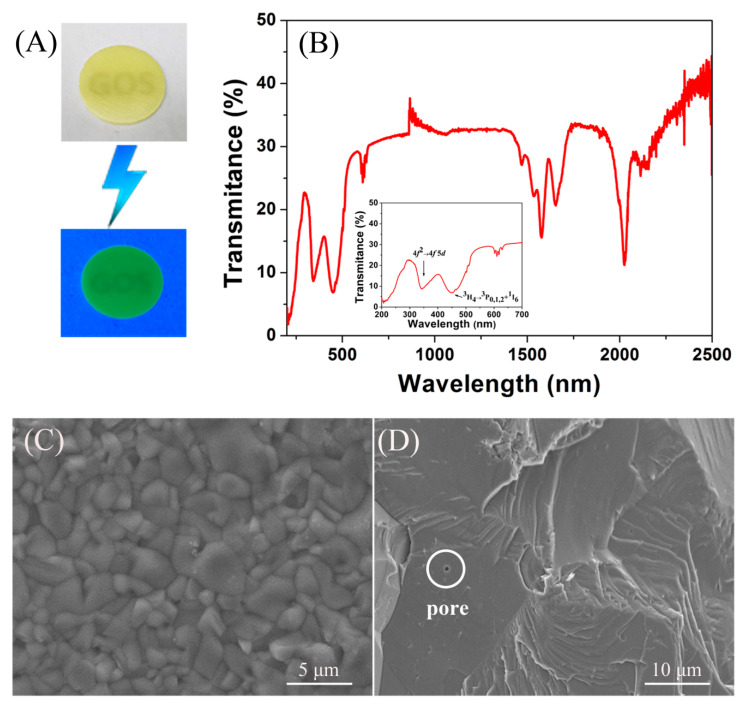
Appearance (**A**) and in-line transmittance (**B**) of the GOS:Pr ceramic scintillator, and SEM micrographs of the surface (**C**) and fracture surface (**D**) of the sintered ceramic. The lower part in panel (**A**) is the appearance of GOS:Pr ceramic under irradiation from a 254 nm UV lamp. The inset in panel (**B**) is the enlargement of its in-line transmittance curve from 200 to 700 nm.

**Figure 4 nanomaterials-10-01639-f004:**
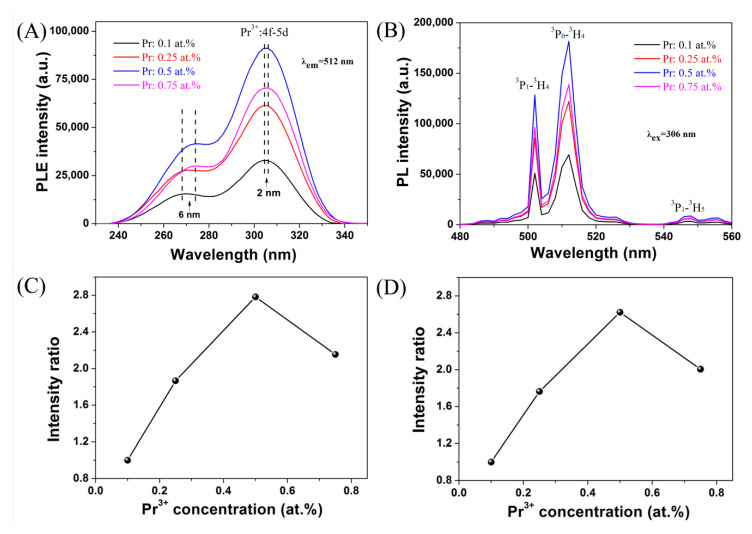
Photoluminescence excitation (PLE) (**A**) and photoluminescence (PL) (**B**) spectra of (Gd_1−*x*_Pr*_x_*)_2_O_2_S (*x* = 0.001–0.0075) phosphor powders, PLE intensities of the 306 nm excitations normalized to 1 for the lowest value (**C**), and PL intensities of the 512 nm emissions normalized to 1 for the lowest value (**D**).

**Figure 5 nanomaterials-10-01639-f005:**
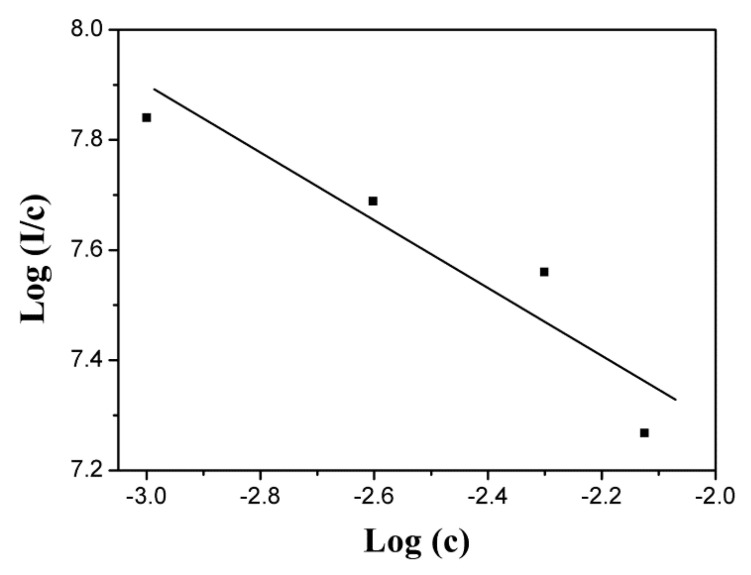
The relationship between *log*(*I*/*c*) and *log*(*c*) for the GOS:Pr phosphors.

**Figure 6 nanomaterials-10-01639-f006:**
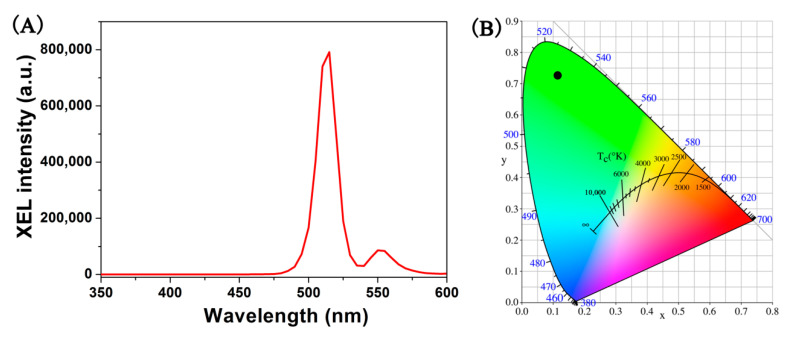
X-ray excited luminescence (XEL) spectrum of the translucent GOS:Pr ceramic scintillator (**A**) and its 1931 CIE chromaticity diagram (**B**).

**Figure 7 nanomaterials-10-01639-f007:**
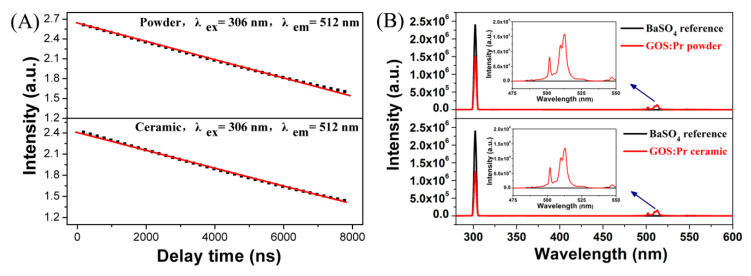
Fluorescence decay behaviors of the GOS:Pr phosphor powder and scintillation ceramic for the 512 nm Pr^3+^ emission under 306 nm excitation (**A**) and their quantum efficiency spectra obtained under 306 nm excitation (**B**).

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
