# Peer review of "Synthesis of Green-Emitting Gd2O2S:Pr3+ Phosphor Nanoparticles and Fabrication of Translucent Gd2O2S:Pr3+ Scintillation Ceramics"

_nanomaterials, 2020, doi:10.3390/nano10091639_

Round 1

Reviewer 1 Report

This paper present the performances of a translucent Gd2O2S:Pr ceramic scintillator with an in-line transmittance of 31% at 512 nm was successfully fabricated by argon-controlled sintering. The authors presented the quantum efficiency and a spectral analysis of the emission spectrum.

In my opinion the paper can be accepted in this status.

Author Response

Thank you for your appreciation on our work.

Reviewer 2 Report

Major issues: The most important observation concerns the physics behind the concentration quenching, Pr-Pr interaction and X-ray luminescence.

(1) the process of concentration quenching is not discussed (cross-relaxation via the intermediary levels, others ?)

(2) the type of Pr-Pr interaction (multipole versus exchange). The authors treat the issue rather superficially, the reference is not relevant perhaps authors refer to the model of direct transfer developed by Inokuti and Hirayama (M. Inokutl and F. Hirayama, J. Chem. Phys. 43, 1978 (1965)) ? And here, fitting the emission decays instead of emission intensities is much more reliable. Besides, comparison of emission intensities is tricky unless some experimental precaution is taken (and it appears this is not the case).

(3) Fitting in Fig. 5 is not convincible (please, see above) . There are only 4 points, errors are not included, if the first three points are selected these fit a line perfectly , etc.

(4) From both Experimental section and Results sections, it is not clear how and what the authors compare as XEOL intensity. Why the authors selected CdWO4 as reference ? The emission shape and widths are significantly different; The factor of ~2.4 times was estimated considering peak intensity or integrated area ?

Minor:

(1) Quantum yield spectra , this is not a familiar term for spectroscopists. I would suggest the authors to provide some references.

(2) The decays should be represented in semi-log scale (Fig. 7).

(3) Reference missing and vague statements : “Wet chemical route is an environmentally friendly way for the synthesis of morphology-controllable oxysulfide particles including hydrothermal reaction, homogenous precipitation, and direct precipitation, among which direct precipitation is preferred for its higher batch yield and simple operation.”; “which obeys Vegard’s law”; “The weak bands at 245‒285 nm are owing to the absorption of host lattice.”; “under high-energy ray excitation, lots of electron-hole pairs in the host lattice are created” and so on.

Author Response

Thank you for your comments. Please refer the attanchmnet for our responds.

Reviewer 3 Report

REVIEW

The paper suggests a route to synthesize a GOS:Pr doped ceramic scintillator. The results are good in terms of scintillating properties and the topic of ceramic scintillators for UV and Xray is interesting and topical.

  • Line 29 The paper has a wide bibliography, but in my opinion the authors have to add some more literature at [1,2]. There are a lot of papers on this topic
  • Line 109 Clarify the origin of the structure of GOS in fig. 1 (elaborated from ref 29, direct calculation by the authors ...)
  • Line 120 As in the paper the Debye-Scherrer formula has a correction factor, please put a reference
  • Lines 140-142 This sentence makes sense if it is related to the size of the sample: a thick ceramic can hardly be translucent. Authors must specify the size of the samples, it is a significant information
  • 3B and lines 144-146  Put an inset to enlarge the lower wavelengths transmittance (till  about 700)
  • 4 for clarity put the excitation and emission wavelength in A and B
  • Line 199 generally in chemistry asterisk underlines an asymmetry, what is the meaning in this case?
  • Lines 213-217 This part is not clear: I think the authors want to speak about the decay time, as stated in literature. The result is not ultrafast (for sure if compared to traditional scintillating crystals), but confrontable with other GOS and CWO (see for instance “Paul Lecoq, Development of new scintillators for medical applications, Nuclear Instruments and Methods in Physics Research A 809 (2016) 130–139). I think that the authors have to clarify and put a comparison table.
  • 7 and line 219 As a matter of fact being “efficiency” in literature is expressed in % not in generic AU. Moreover, the BaSo4 reference is the same of reference [35], can explain this choice?
  • Line 230 231 In conclusion better to avoid acronyms: I suggest cold isostatic pressing instead of CIP and Gd2O2S:Pr (GOS:Pr)
  • Line 238 As explained before (8) the “ultrafast” is not absolute but referred to some specific samples. In my opinion is better “faster than” or related to specific applications. Please explain and modify.
  • Did the authors test the stopping power? Do they refer to the literature on similar samples?
  • Some English improvements (line 52 sintegirngà sintering….)

Author Response

(The authors gave the same response as above.)

Reviewer 4 Report

The authors report the development of GOS:Pr translucent ceramic scintillators using a relatively low temperature sintering. GOS:Pr ceramics have long been used as efficient scintillators, and the novelty of this paper is on the synthesis method. The structure of the ceramics is extensively characterized, and the optical and scintillation properties are characterized in detail. This paper is interesting for readers whose area of expertise is radiation detection and phosphors. I recommend the publication of this paper after some minor modifications on the unclear points in the present manuscript listed below.

  1. Concentrations of metal ions in the mother liquor are missing. Also, rotational speed during the centrifugation would be helpful for readers.
  2. Taking into account of the difference in the ionic radii of Gd3+ and Pr3+, the lattice parameters are expected to increase with the Pr concentration, as the authors explain in the 1st paragraph of Results and Discussion. Actually, the authors mention that "All the diffraction peaks gradually shift towards low angle sides along with increasing Pr3+ addition, implying Pr3+ doping induces a lattice expansion due to the a little larger ionic radius of Pr3+ than that of Gd3+" in the same paragraph. However, the lattice parameters shown in Figure 1 (c) and (d) decrease with the Pr concentration. The explanation on this point is inconsistent. Also, in the same paragraph, "because the small difference in ionic radii" should be revised to "because of the small difference in ionic radii".
  3. The sintered samples seem to be translucent judged from the photograph of Figure 3 (a). How was the thickness of the sintered samples? It is an important parameter to consider the transparency of the samples.
  4. In the last paragraph of Results and Discussion, the quantum yields of photoluminescence of the samples were presented. On the contrary, the photoluminescence intensity in arbitrary unit is used for discussion in the 4th and 5th paragraphs. The discussion on the intensity in the 5th paragraph should be carefully treated: the concentration quenching should be discussed on the basis of quantum yield. I'm afraid that the photoluminescence intensity dependent on the Pr concentration reflects the fraction of the absorbed excitation light, which is also considered to depend on the Pr concentration. I strongly recommend that the authors discuss the concentration quenching on the basis of quantum yield.
  5. As for the XEL mechanism, the proposed one in this paper is a so-called sequential charge trapping process. An additional process is resonant energy transfer from the excited states in the host, which are generated via recombination of electron-hole pairs. If the authors claim that the former process is dominant in GOS:Pr, some additional evidence or references would be necessary.
  6. The authors compare the scintillation efficiency of their samples and that of commercially available crystal of CdWO4 on the basis of XEL spectra. To provide a correct comparison, they should consider the following points and include them in the manuscript: (i) the sizes of the samples and the reference crystals, (ii) the difference in the absorbed energy of X-rays was considered. (iii) whether the spectra was corrected for the wavelength-dependent sensitivity of the measurement system (wavelength-dependent transmittance of the monochromator and the sensitivity of the photomultiplier tube), (iv) the possible difference in the collection efficiencies of scintillation was considered. In many cases, the estimation of relative scintillation efficiency is difficult on the basis of XEL spectra because of many correction factors needed.
  7. The term "ultrafast" for the photoluminescence decay on the order of microseconds is not suitable for scintillators. As you know, many scintillators have decay time constants much less than microseconds.

Author Response

(The authors gave the same response as above.)

Round 2

Reviewer 2 Report

The authors ' s response is fully unsatisfactory. Therefore, I am keeoping my initial decision (reject). 

Author Response

    The main controversy lies in our method for the investigation of concentration quenching mechanism. In this work, we employ the relative theory developed by Huang (ref. 42), which agrees with Dai and Meng et al (refs. 43 and 44). In fact, this theory formula was deduced according to your recommended luminescence model given by Inokuti and Hirayama. The former has become a facile method to explore the mutual interaction type of luminescence quenching in a solid phosphor and the validity has been demonstrated by lots of previous reports (e.g. F. Ou-Yang, B. Tang, Rare Metal Mater. Eng. 32, 522, 2003; D. Li, S. Lu, H. Wang, et al. Huang, Chin. J. Lumin. 22, 227, 2001; Q. Meng, B Chen, W Xu, et al. J. Appl. Phys. 102, 093505, 2007; J.K. Li, J.-G. Li, Z.J. Zhang, Sci. Technol. Adv. Mater. 13, 035007, 2012; J.K. Li, J.-G. Li, S.H. Liu, et al. J. Mater. Chem. C, 1, 7614, 2013……).

    We supplemented a statement (lines 182‒183) and a typical reference for readers’ better understanding (ref. 44).
